# Sympathetic Burden Measured Through a Chest-Worn Sensor Correlates with Spatiotemporal Gait Performances and Global Cognition in Parkinson’s Disease

**DOI:** 10.3390/s25185756

**Published:** 2025-09-16

**Authors:** Gabriele Sergi, Ziv Yekutieli, Mario Meloni, Edoardo Bianchini, Giorgio Vivacqua, Vincenzo Di Lazzaro, Massimo Marano

**Affiliations:** 1Fondazione Policlinico Universitario Campus Bio-Medico, Viale Alvaro del Portillo 200, 00128 Rome, Italy; gabriele.sergi@unicampus.it (G.S.); e.bianchini@policlinicocampus.it (E.B.); v.dilazzaro@policlinicocampus.it (V.D.L.); 2Research Unit of Geriatrics, Department of Medicine and Surgery, Università Campus Bio-Medico Di Roma, 00128 Rome, Italy; 3Mon4t, Tel Aviv 6706057, Israel; ziv@mon4t.com; 4Neurology Unit, Azienda Ospedaliera Universitaria di Cagliari, 09123 Cagliari, Italy; ma.meloni@aoucagliari.it; 5Department of Human Neurosciences, Sapienza University of Rome, 00185 Rome, Italy; 6AGEIS, Université Grenoble Alpes, 38000 Grenoble, France; 7Laboratory of Microscopic and Ultrastructural Anatomy, Università Campus Bio-Medico di Roma, Via Alvaro del Portillo 21, 00128 Roma, Italy; g.vivacqua@unicampus.it; 8Research Unit of Neurology, Neurophysiology, Neurobiology and Psychiatry, Department of Medicine, Università Campus Bio-Medico di Roma, Via Alvaro del Portillo 200, 00128 Rome, Italy

**Keywords:** heart rate variability, Baevsky stress index, autonomic dysfunction, gait analysis, cognitive function

## Abstract

**Highlights:**

**What are the main findings?**
Time-domain heart rate variability parameters are significantly correlated with spatiotemporal gait features in Parkinson’s disease.The Stress Index, a measure of sympathetic activity, is associated with poorer gait performance and reduced cognitive function.

**What is the implication of the main finding?**
Measures derived by heart rate variability may serve as non-invasive biomarkers for the evaluation of functional mobility and cognition in Parkinson’s disease.Heart rate variability and autonomic functions should be targets for future clinical trials evaluating motor performances.

**Abstract:**

Autonomic dysfunction is a key non-motor feature of Parkinson’s disease (PD) and may influence motor performance, particularly gait. While heart rate variability (HRV) has been associated with freezing of gait, its relationship with broader gait parameters remains unclear. The objective was to investigate correlations between resting-state HRV time-domain measures and spatiotemporal gait parameters during comfortable and fast walking in patients with idiopathic PD. Twenty-eight PD patients (mean age 68 ± 9 years) were evaluated at Campus Bio-Medico University Hospital. HRV was recorded at rest using the e-Sense pule™ portable sensor, including the Baevsky’s Stress Index a measure increasing with sympathetic burden. Gait parameters were assessed via the 10 m Timed Up and Go (TUG) test using the Mon4t™ smartphone app at comfortable and fast pace. Clinical data included UPDRS III, MoCA, and disease characteristics. Gait metrics significantly changed between walking conditions. HRV parameters clustered separately from gait metrics but intersected with significant correlations. Higher Stress Index values, reflecting sympathetic dominance, were associated with poorer gait performance, including prolonged transition times, shorter steps, and increased variability (*p* < 0.001, r = 0.57–0.61). MoCA scores inversely correlated with the Stress Index (r = −0.52, *p* = 0.004), linking cognitive and autonomic status. UPDRS III and MoCA were related to TUG metrics but not HRV. Time-domain HRV measures, particularly the Stress Index, are significantly associated with spatiotemporal gait features in PD, independent of gait speed. These findings suggest that impaired autonomic regulation contributes to functional mobility deficits in PD and supports the role of HRV as a biomarker in motor assessment.

## 1. Introduction

Parkinson’s disease (PD) is the second most common neurodegenerative disorder after Alzheimer’s disease, affecting approximately 1–2% of individuals over the age of 60 [1,2]. It is the fastest-growing neurological condition worldwide, largely due to its multifactorial pathogenesis, in which environmental neurotoxins and aging play significant roles [3]. The hallmark motor symptoms of PD (i.e., bradykinesia, rigidity, and tremor) are caused by striatal dopamine depletion resulting from progressive neuronal loss in the substantia nigra pars compacta and the nigrostriatal pathway [1]. Gait and posture disturbances in PD are common motor issues presenting aside key motor features and have a complex pathophysiology, involving both dopaminergic and non-dopaminergic mechanisms that differentially affect various aspects of human walking [4]. A link between gait performance and autonomic nervous system function has been proposed, with autonomic dysfunction being recognized as a key non-motor feature of PD [5]. Although the pathophysiology of autonomic dysregulation in PD is not fully understood, it is thought to involve both peripheral and brainstem neuronal pathology [6]. There is a partial overlap between the neural networks governing autonomic functions and those involved in locomotion [7]. For example, gait disturbances such as freezing of gait (FOG) have been linked to orthostatic hypotension [5] and reduced heart rate variability (HRV) [8]. HRV reflects fluctuations in the time intervals between consecutive heartbeats, enabling physiological adaptation to environmental demands. A healthy heart exhibits constant variability, adapting rapidly to both physical and psychological stimuli [9]. While associations between HRV and FOG have been documented, previous studies have shown only partial correlations between HRV and other gait parameters [10].

Recent preliminary evidence highlights the utility of Baevsky’s Stress Index as a synthetic marker of autonomic modulation, with higher values indicating greater psychophysiological stress. In comparative studies, both PD and Multisystem Atrophy (MSA) patients displayed significantly elevated Stress Index together with other indexes of sympathetic activity (i.e., SNSi) compared with healthy controls, while parasympathetic activity (i.e., measured through the PNSi) was reduced (Brisinda et al., 2024 [11]). More interestingly, recent longitudinal evidence indicates that altered autonomic cardiovascular responses may precede the clinical onset of PD and in a large cohort study, van Duijvenboden et al. [12] reported that abnormal heart rate profiles (i.e., heart rate recovery) during exercise were associated with an increased risk of incident PD, supporting the role of early autonomic dysfunction as a prodromal marker of the disease. All together, these findings suggest that synucleopathies are characterized by impaired autonomic balance and heightened stress burden, possibly even before disease diagnosis.

In this small pilot study, we aimed to explore the correlation between time-domain HRV measures—reflecting parasympathetic and sympathetic activity—and spatiotemporal gait parameters in patients with PD. Additionally, we examined how resting HRV re-sponds to two different walking conditions: comfortable and fast pace. This study contributes to the field by integrating HRV-derived autonomic indices with smartphone-based gait analysis to provide a multidimensional characterization of PD. To our knowledge, this is among the first attempts to combine autonomic and motor features within a sensor-based framework [10], with potential implications for digital health monitoring and personalized disease management.

## 2. Materials and Methods

### 2.1. Patient Selection and Clinical Evaluation

Twenty-eight subjects were included in this study. Patients were consecutively enrolled at the movement disorder clinic of Fondazione Policlinico Universitario Campus Bio-Medico of Rome as affected by idiopathic PD. Exclusion criteria were: Hoehn and Yahr >3, inability to walk unassisted or to adapt their own walk to a faster pace, known cardiac arrhythmias or chronic heart diseases, therapy with antiarrhythmics or beta blockers for hypertension or tremor, and being treated with deep brain stimulation or other device aided therapies. Patients with dementia, neuro-logical comorbidities, neurovascular or other major cardiovascular diseases were also excluded. Only patients on levodopa and a stable dopaminergic therapy (i.e., no therapeutic changes in the previous 4 weeks) were enrolled. Of 50 consecutively screened patients, 28 matched the enrollment criteria and were included in the study. Demo-graphic and disease specific data were collected, including age, sex, disease duration, Unified Parkinson’s Disease Rating Scale (UPDRS) part III and modified Hoehn and Yahr scale, levodopa equivalent daily doses (LEDD) and freezing of gait questionnaire (FOG-Q). The Montreal Cognitive Assessment (MoCA) scale was used as a measure for global cognition.

### 2.2. Sensor-Based Measurement—Heart Rate Variability

Heart rate variability (HRV) was assessed using the eSense pulse™ device (Mindfield Biosystems, Gronau, Germany; www.mindfield-esense.com), a wearable single-lead ECG sensor integrated into an elastic chest strap. The device records raw cardiac signals with an internal sampling frequency of 500 Hz and transmits R–R intervals at 5 Hz via Bluetooth Low Energy. It detects heart rates in the range of 30–240 beats per minute with an accuracy of ±2 bpm. The strap electrodes are composed of conductive silicone embedded in a thermoplastic polyurethane support to ensure stable skin contact. The system is specified to operate under ambient conditions of 5–40 °C and relative humidity up to 95%.

Recordings were conducted while participants were standing at rest for a duration of 2 min [8]. The extracted HRV metrics included mean, minimum, maximum, and delta heart rate (HR) values, along with time-domain parameters such as mean heart rate (HR), mean RR interval, standard deviation of NN intervals (SDNN), root mean square of successive differences (RMSSD), and Baevsky’s Stress Index.

The Baevsky’s Stress Index (SI) is a time-domain parameter of HRV that reflects the degree of sympathetic activation. It is derived from the statistical distribution of consecutive RR intervals and is calculated as:$$SI=\frac{AMo\times 100}{2\times Mo\times MxDMn}$$
where AMo is the amplitude of the mode (percentage of NN intervals corresponding to the most frequent RR value), Mo is the modal value of the RR intervals, and MxDMn is the range between the longest and shortest RR intervals. Higher values indicate reduced variability and greater sympathetic dominance, while lower values reflect higher autonomic flexibility. In healthy adults, normative values are typically 50–150 units, with sustained elevations suggesting autonomic imbalance and increased cardiovascular risk [13]. Frequency-domain parameters were not included in order to preserve statistical power, given the pilot and exploratory design of the study, and because of the high susceptibility of these measures to artifacts (e.g., tremor in PD) that could bias their interpretation.

Data acquisition was managed through the dedicated eSense v7.2.4 application, which allows export in CSV or PDF format for offline statistical analysis.

### 2.3. Sensor-Based Measurement—Spatiotemporal Gait Measures

Gait performance was evaluated using the Mon4t™ smartphone application (Tel Aviv, Israel; www.mon4t.com), installed on a Samsung Galaxy S5 device (Samsung, Seoul, Republic of Korea). The Galaxy S5 is equipped with a triaxial accelerometer and a triaxial gyroscope, both based on microelectromechanical systems (MEMS) technology. The accelerometer measures linear acceleration within a range of approximately ±2 g to ±16 g with a resolution in the milli-g scale, enabling detection of body displacement along the three spatial axes. The gyroscope measures angular velocity up to ±2000°/s with sub-degree resolution, allowing accurate capture of rotational movements and postural transitions. These sensors stream continuous raw data at a sampling frequency of up to 200 Hz, which the Mon4t™ application processes to extract clinically relevant gait parameters. During the 10 m Timed Up and Go (TUG) test, each participant performed three trials under two conditions: comfortable walking speed and self-selected fast walking speed. The following spatiotemporal parameters were recorded: stand-up time, walking time, rotation time, sit-down time, total TUG time, step length, step-to-step correlation, and number of steps. The Mon4t™ application has FDA clearance, and its use has been extensively validated in multiple clinical trials, supporting the reliability of smartphone-based inertial sensing for gait analysis in Parkinson’s disease [14,15,16,17,18].

All clinical and sensor-based assessments were conducted during the patient’s defined ON-medication state, in a well-lit environment, and at a consistent time of day to minimize external variability.

### 2.4. Statistical Analysis for Descriptive Statistics and Correlation

Data are described as mean ± standard deviation (SD). Data distribution has been tested through the Shapiro-Wilks test. Changes in time of variables were tested through ANOVA methods.

Correlations between variables have been carried out through the Pearson test in a single matrix and significant relationships were corrected for multiple comparisons through the False Discovery Rate (FDR) method of Benjamini–Hochberg. This analysis was carried out across 3 cluster of data: demographic/clinical data (age, sex, UPDRS III, Hoehn and Yahr scale, FOG-Q, MoCA, LEDD), HRV (HRV max, HRV min, HRV delta, HRV mean, SDNN, RMSSD, Stress Index, Mean HR, Mean RR) data and spatiotemporal gait data extracted by TUG test (Stand Up Time, Walk Away Time, Rotation Time, Walk Back Time, Sit Down Time, Total Time, Rotation steps, Step correlation, Step frequency, Sway, Step number, Step length) at comfortable and fast walking.

Statistics were performed through the JMP 18.0 software (SAS Inc., Cary, NC, USA). A *p*-value < 0.05 was adopted for statistical significance.

The study was conducted according to the Helsinki declaration of human rights and has been approved by the Ethical committee of the University Campus Bio-Medico di Roma (03.21 OSS ComEt CBM). All subjects signed a written informed consent.

## 3. Results

Our sample included 28 adults (mean age 68 ± 9), with idiopathic PD featured by a bilateral mild-to-moderate motor disease overall (mean modified Hoehn and Yahr 2 ± 0.5 and UPDRS III was 18 ± 6.8) and a disease duration from symptom onset of 6.9 ± 5.9 years. A higher prevalence of male subjects (75%) was found, as expected (Table 1). Changes between comfortable speed vs. fast speed TUG values showed reduced performance time (total time −3.7%), except for sit down time, increased step length (+3.4%) and reduced walking steps (−3%), except for rotation, without any change in sway (0%)—as expected by a cautious increase in walking speed. Data from resting-state HRV recordings and the 10 m TUG tests, performed at both comfortable and fast walking speeds, are summarized in Table 1. Patients’ gait parameters significantly changed during normal and fast walking, except for sit down time, rotation steps, step frequency and postural sway. The correlation analysis between clinical data, HRV and spatiotemporal gait parameters is shown in Figure 1 and Table 2. Two distinct clusters of variables emerged: one comprising HRV measures, and the other including TUG-derived gait features. The intersection between these two clusters revealed significant correlations, suggesting a relationship between clinical parameters, autonomic regulation and physical mobility. Specifically, HRV min and Stress Index related to spatiotemporal gait parameters at both comfortable and fast-paced TUG. Both, respectively, related to worse performances at TUG test supported by stand up (*p* = 0.002, r = 0.57; *p* < 0.001, r = 0.69), walk away (*p* = 0.009, r = 0.49; *p* < 0.001, r = 0.69), rotation (*p* = 0.019, r = 0.44; *p* = 0.001, r = 0.71), walk back (*p* = 0.006, r = 0.51; *p* < 0.001, r = 0.70), sit down (*p* = 0.021, r = 0.43, *p* = 0.001, r = 0.57) and total TUG time ( *p* = 0.005, r = 0.51; *p* < 0.001, r = 0.70), with only Stress Index showing an additional direct relationship with step number (*p* = 0.001, r = −0.52) and an inverse relationship with step length (*p* = 0.005, r 0.58) and step correlation (*p* = 0.005, r = −0.52) (Figure 1). Higher HRV minimum values and elevated Stress Index were associated with poorer gait performance, reflected by longer execution times. Moreover, higher Stress Index values were linked to a greater number of shorter steps and increased variability in walking performance. Correlation parameters were similar across TUG modalities, with only sit down time showing no relationship with HRV min at fast TUG. Scatterplot diagrams and exact *p*-values are reported in Appendix A.

As expected, clinical parameters of motor state, cognition and gait showed a relationship with spatiotemporal TUG test gait features, while disease duration and LEDD did not (Table 2). Specifically, UPDRS III and MoCA scores were significantly associated with TUG body transitions (*p* < 0.001, r = 0.49–0.61 for both variables), overall TUG time (*p* < 0.001, r = 0.68 for UPDRS III and r = −0.59 for MoCA), rotations (*p* < 0.001, r = 0.55 for UPDRS III), and step variability (*p* < 0.05, r = −0.46 for UPDRS III and r = 0.51 for MoCA) (Table 2). These findings indicate that greater motor impairment (higher UPDRS III) and poorer cognitive function (lower MoCA) were both related to worse gait features overall. Similar correlations were observed between clinical parameters and fast TUG outcomes. Exact *p*-values are provided in Appendix A. Conversely, clinical motor parameters showed no association with HRV data, but MoCA scores were inversely correlated with the Stress Index, indicating that better cognitive performance was observed in patients with lower stress levels (r = −0.52, *p* = 0.004).

## 4. Discussion

In this study, we documented the presence of a relationship between specific heart rate variability (HRV) measures and gait parameters in a well-characterized cohort of patients with intermediate-stage PD in different walking conditions (e.g., comfortable vs. fast pace). The fast-paced condition at the 10-meter TUG test elicited expected small improvements in gait performance, including reduced execution time, fewer steps, and increased step length (Table 1). However, sway did not increase as would typically be observed in healthy subjects when step length increases, suggesting that patients with PD adopt a more cautious approach to walking [18].

While HRV and gait disturbances have been examined independently in previous research, a direct association between these domains—independent of freezing of gait—has not been previously established. Freezing of gait has already been associated with autonomic dysregulation [19], and Heimler and colleagues further documented reduced HRV in patients experiencing freezing of gait, regardless of their dopaminergic treatment status [8].

Our pilot study is the first to explore the role of the Stress Index—a marker of cardiac sympathetic activity [20]—on gait performance, suggesting that elevated Stress Index values are associated with PD-specific gait alterations across various walking conditions (e.g., comfortable vs. fast pace).

Interestingly, Stress Index has been investigated as a potential biomarker in differentiating PD by other synucleopathies (e.g., MSA). However, the absence of significant differences between these groups suggested that Stress Index alone does not provide sufficient discriminatory power for early differential diagnosis. Nevertheless, its positive correlation with other HRV biomarkers of sympathetic activity (i.e., the SNSi) and motor performances, and the feasibility of remote, telematic assessment of HRV support the potential of Stress Index and other HRV indexes as a practical biomarker for monitoring autonomic dysfunction and sympathetic burden in neurodegenerative conditions such as PD [11].

The observed associations between HRV and TUG test variables highlight a critical interplay between autonomic nervous system regulation and motor performance. Two key findings emerge: the sympathetic overdrive, supported by lower HRV and higher Stress Index values, is correlated with poorer motor outcomes on the TUG, including prolonged times for standing up, gait initiation, and turning. These results suggest that heightened autonomic stress may impair movement efficiency and compromise postural control. An additional noteworthy finding is the inverse relationship between Stress Index and MoCA scores. The MoCA test, a broad measure of global cognitive function, was significantly associated with several spatiotemporal gait parameters. Cognitive dysfunction is a well-established contributor to gait impairment in PD [21], likely due to overlapping neural substrates such as the basal forebrain cholinergic system and prefrontal dopaminergic networks [22]. The observed association between sympathetic overdrive and cognitive function is consistent with evidence on the role of autonomic balance in cognition. The sympathetic nervous system has been linked to cognitive performance, with moderate activation supporting the physiological arousal required for optimal functioning [23,24]. By contrast, impaired sympathetic activity has been associated with dementia in PD and with memory deficits following pharmacological suppression [25,26]. Over longer periods, chronic sympathetic overactivation may contribute to maladaptive neural and vascular changes detrimental to cognitive aging [27,28]. Taken together, these findings suggest that moderate SNS activity may facilitate cognition, whereas sustained sympathetic dominance exerts harmful effects. Moreover, our findings are consistent with previous findings showing that autonomic modulation differs across PD motor subtypes, with reduced HRV and sympathetic predominance in akinetic-rigid compared to tremor-dominant patients [29], our results further support the role of autonomic dysfunction as a contributor to the heterogeneity of clinical manifestations in PD.

Non-motor symptoms may precede motor signs by several years, as seen in hyposmia, REM sleep behavior disorder, and constipation. Autonomic system impairment is a core non-motor complication of PD, commonly presenting as dysautonomia [6]. This includes impaired cardiovascular regulation, orthostatic hypotension, supine hypertension, sudomotor dysfunction, constipation, urinary urgency, and nocturia.

HRV is a non-invasive measure of cardiac autonomic nervous system function, reflecting the dynamic balance between sympathetic and parasympathetic input to the heart. Time-domain parameters such as SDNN (standard deviation of NN intervals) and RMSSD (root mean square of successive differences) provide insights into overall HRV and parasympathetic activity, respectively. RMSSD is sensitive to short-term vagal modulation. In addition, the Stress Index, also known as Baevsky’s Stress Index, quantifies the sympathetic influence on heart rate by analyzing the distribution of RR intervals; higher Stress Index values reflect increased sympathetic tone and reduced variability, often associated with physiological stress or autonomic imbalance [9]. Hence, elevated Stress Index values are typically observed under conditions of increased sympathetic drive or reduced parasympathetic modulation, whereas low Stress Index values indicate reduced sympathetic activity. Heart rate variability (HRV) alterations in PD reflect neurodegeneration within central autonomic pathways, including brainstem nuclei such as the dorsal motor nucleus of the vagus, locus coeruleus, and medullary cardiovascular centers, resulting in impaired parasympathetic function. Additionally, dysfunction of basal ganglia and limbic circuits further compromises autonomic and stress regulation. Reduced baroreflex sensitivity and vagal tone contribute to cardiovascular instability, manifesting as blood pressure variability and orthostatic hypotension [6]. HRV reductions correlate with disease severity, supporting its potential as a biomarker of autonomic dysfunction and disease progression [30]. Notably, exercise-based interventions have been shown to enhance both motor performance and autonomic function in PD patients, as evidenced by favorable changes in HRV parameters. This parallel effect further reinforces the association between motor impairment and autonomic dysfunction, suggesting that interventions targeting autonomic regulation may promote functional mobility through shared neurophysiological mechanisms [31].

Together, these metrics offer a multidimensional view of autonomic function, with potential relevance in characterizing motor and non-motor features of PD. The observed association between reduced HRV and prolonged TUG times supports the notion that reduced autonomic flexibility is linked to impaired physical performance in PD. The Stress Index was also negatively associated with motor performance. Higher Stress Index values were correlated not only with longer task duration but also with shorter step length, increased step count, and greater step variability. This suggests that individuals with greater physiological stress require more effort to initiate and execute gait sequences, even under different walking demands. Notably, both HRV and Stress Index affected walking parameters independently of the walking condition.

Indeed, interventions targeting the autonomic system—such as vagus nerve stimulation or even exercise-based therapies—are under investigation and show promising effects on cognition, gait, and freezing of gait [32,33]. HRV is a validated biomarker of autonomic function, and growing evidence supports its relevance in the regulation of locomotor control. For instance, subclinical neurogenic orthostatic hypotension has been associated with slower gait speed, shorter stride length, prolonged postural transitions, and increased sway [5], reinforcing the clinical utility of early autonomic assessment in PD.

Despite this, HRV is not routinely evaluated in clinical practice for patients with PD. Other autonomic measures—such as orthostatic blood pressure testing—are commonly performed even in outpatient settings. Autonomic dysfunction is central to the non-motor spectrum of PD and is also crucial for differentiating idiopathic PD from atypical parkinsonian syndromes, such as MSA. Notably, cardiac (123)I-MIBG scintigraphy, which aids in distinguishing PD from MSA, has been linked to HRV, with reduced myocardial uptake correlating with diminished HRV in PD [34].

This study has several limitations beyond the small sample size. Most notably, the lack of a controlled study design limits our ability to determine whether the observed associations are specific to PD. Furthermore, the absence of a prospective assessment constrains our capacity to draw conclusions about the longitudinal relationship between HRV measures and disease progression, which in this context can only be inferred indirectly through clinical indicators of motor severity. Low numbers prevent us from speculating on a possible gender effect that would be considered in future studies given the pathophysiologic relevance in autonomic control [35]. Indeed, the predominance of male participants, which in our sample exceeded the expected male-to-female ratio of approximately 2:1 typically reported in Italian PD epidemiology, may constitute a relevant source of bias. Sex differences are increasingly recognized as critical modifiers of PD, influencing not only the clinical motor phenotype but also cognitive trajectories and autonomic regulation [36]. Women with PD have been reported to show different patterns of motor progression, a relatively higher burden of non-motor symptoms such as anxiety or fatigue [37,38], and a distinct sensitivity to dopaminergic therapies, particularly in terms of cardiovascular and autonomic effects [39]. As our study focused on cognitive performance, spatiotemporal gait parameters, and heart rate variability, all domains known to be modulated by sex-related factors, the unbalanced sex distribution may have influenced the overall outcomes. Future studies should therefore aim for sex-stratified analyses or balanced recruitment to disentangle the contribution of sex to these clinical and physiological endpoints. The observed interplay between MoCA scores, the Stress Index, and gait features is particularly compelling and warrants further investigation using domain-specific cognitive assessments. Lastly, we did not include HRV parameters derived from frequency-domain analysis (e.g., low frequencies—LF, high frequencies—HF, and LF/HF ratio), relying solely on time-domain metrics [9]. Indeed, frequency-domain metrics have been already associated with postural control and disease phenotype in PD [26], showing that a blunted cardiac autonomic function in both the supine and orthostatic positions is associated with worse postural control and akinetic-rigid phenotype [29]. While frequency-domain parameters can provide complementary insights, particularly regarding sympathetic–parasympathetic balance, in this pilot study we opted for time-domain measures due to their robustness, simplicity, and reduced risk of type I error in a small cohort. Moreover, frequency-domain measures require stronger assumptions about signal stationarity, which may not always be satisfied in short-term or real-life recordings in PD being more sensitive to noise (e.g., presence of tremor). We acknowledge that this choice may have underestimated the strength of autonomic–motor associations, and therefore highlight the importance of including both domains in future studies with larger, well-powered samples.

Finally, our sample was tested during a definite levodopa ON condition and cannot capture the variability of autonomic response to levodopa plasmatic fluctuations as suggested by previous studies on levodopa response [40]. This may further limit the generalizability of our findings, as they may not reflect what could be observed in levodopa-naïve patients, in patients experiencing motor fluctuations, or during continuous telemonitoring of autonomic parameters, where the influence of levodopa and other dopaminergic therapies may play a prominent role in autonomic control. Peripheral dopamine metabolism interacts with autonomic function at multiple levels [39].

Overall, this analysis reveals significant associations between HRV metrics and TUG performance, emphasizing the role of autonomic regulation in functional mobility. These findings may inform the development of strategies aimed at improving motor outcomes by enhancing autonomic stability. Moreover, our findings also provide novel insights that advance the sensing field. By integrating autonomic indices derived from portable HRV devices, such as the Stress Index, with smartphone-based gait analysis, we demonstrate the feasibility of a multidimensional characterization of PD that simultaneously captures motor and autonomic dysfunction. This approach has important implications for sensor-based monitoring systems, suggesting that relatively low-cost and widely available devices can be leveraged for continuous assessment of disease progression and treatment effects. Beyond clinical research, these results open the way to novel applications in real-world gait assessment, where smartphone-embedded inertial sensors may enable remote and ecologically valid monitoring of mobility and autonomic stress burden [Marano]. Given the pilot nature of this study, future research should aim to validate these findings in larger and more diverse cohorts, ideally including longitudinal follow-up and neuroimaging/neurophysiology studies [41] to determine the predictive value of autonomic and gait parameters for mobility decline. Ultimately, such integration into digital health technologies could support personalized care, early detection of clinical changes, and timely therapeutic adaptation.

## Figures and Tables

**Figure 1 sensors-25-05756-f001:**
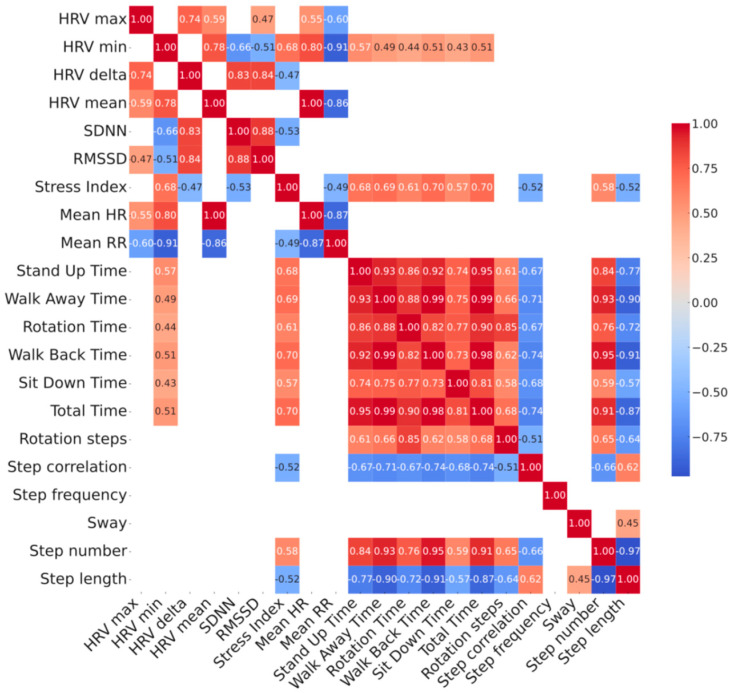
Correlation heatmap between baseline HRV parameters vs. TUG parameters measured at comfortable (top panel) and at fast (bottom panel) pace in 28 patients with Parkinson’s disease. Relationship between HRV and gait parameters has been shown in the left bottom quadrant of each heatmap. Color map reporting the strength of correlation with color scales of the heatmaps represent the correlation coefficient which is also embedded numerically inside each intersection. Only *p* < 0.05 corrected with the FDR method are shown. See Appendix A for complete list of *p*-values. Stress Index, Baevsky’s Stress Index; FDR, False Discovery Rate; FOG-Q, Freezing Of Gait Questionnaire; HF, High frequency; HR, Heart Rate; HRV, Heart Rate Variability; LEDD, Levodopa Equivalent Daily Dosages; LF, Low frequency; MoCA, Montreal Cognitive Assessment; PD, Parkinson’s Disease; RMSSD, Root Mean Square of Successive Differences; RR, R-R interval; SDNN, Standard deviation of NN intervals; TUG, Timed up and go test; UPDRS, Unified Parkinson’s Disease Rating Scale.

**Table 1 sensors-25-05756-t001:** Description of the Parkinson’s disease population (*n* = 28) population in study, including demographical variables, disease specific parameters, heart rate variability parameters and TUG spatiotemporal gait measures. Comfortable pace vs. fast TUG values have been compared with Student *t* test and results reported under the *p*-value column. Comparison shows changes in walking parameters according to the different task.

Variable		Mean/*n* (%)	sd			*p*-Value
Demographic and disease specific	Sex (M)	21 (75%)				
	Age	68.357	9.121			-
	Disease duration	6.954	5.924			-
	Hoehn & Yahr	2.000	0.451			-
	MoCA	23.643	2.147			-
	UPDRS III	18.964	6.818			-
	LEDD	757.750	352.559			-
	FOG-Q	4.893	4.131			-
Heart rate variability	HRV max	85.314	16.772			-
	HRV min	68.354	12.202			-
	HRV delta	16.961	16.965			-
	HRV mean	75.898	11.780			-
	SDNN	39.796	41.339			-
	RMSSD	31.942	50.036			-
	Stress Index	522.041	575.010			-
	Mean HR	75.564	11.833			-
	Mean RR	821.033	127.584			-
		Normal		Fast		
TUG parameters	Stand Up Time (s)	1.113	0.366	1.077	0.366	0.029
	Walk Away Time (s)	4.159	1.464	3.946	1.467	0.001
	Rotation Time (s)	1.115	0.450	1.093	0.452	0.028
	Walk Back Time (s)	4.264	1.485	4.094	1.455	0.001
	Sit Down Time (s)	2.126	0.501	2.094	0.570	0.226
	Total Time (s)	12.776	4.071	12.303	4.085	0.001
	Rotation steps (*n*)	3.586	0.663	3.583	0.728	0.940
	Step correlation (R2)	0.536	0.178	0.556	0.187	0.011
	Step frequency	1.697	0.095	1.690	0.111	0.115
	Sway (m)	0.012	0.003	0.012	0.003	0.881
	Step number	15.767	4.143	15.290	4.136	0.001
	Step length (m)	0.203	0.045	0.210	0.049	0.001

Stress Index, Baevsky’s Stress Index; FDR, False Discovery Rate; FOG-Q, Freezing Of Gait Questionnaire; HR, Heart Rate; HRV, Heart Rate Variability; LEDD, Levodopa Equivalent Daily Dosages; MoCA, Montreal Cognitive Assessment; RMSSD, Root Mean Square of Successive Differences; RR, R-R interval; SDNN, Standard deviation of NN intervals; TUG, Timed up and go test; UPDRS, Unified Parkinson’s Disease Rating Scale.

**Table 2 sensors-25-05756-t002:** Correlation between baseline clinical disease specific and TUG gait parameters at comfortable pace corrected for multiple comparisons.

Variable	Age	Dis. Duration	Hoehn & Yahr	MoCA	UPDRS III	LEDD	FOG-Q	Stand Up Time	Walk Away Time	Rotation Time	Walk Back Time	Sit Down Time	Total Time	Rotation Steps	Step Correlation	Step Frequency	Sway	Step Number	Step Length
Age	1.00 **	0.24	−0.03	−0.30	0.30	−0.00	−0.17	0.21	0.28	0.22	0.30	0.27	0.29	0.24	−0.21	0.25	−0.38	0.35	−0.43
Disease duration	0.24	1.00 **	0.33	−0.10	0.35	0.40	0.08	0.25	0.29	0.31	0.31	0.22	0.30	0.38	−0.17	−0.26	−0.42	0.31	−0.28
Hoehn & Yahr	−0.03	0.33	1.00 **	−0.21	0.51 *	0.13	0.29	0.30	0.33	0.32	0.34	0.19	0.33	0.42	−0.39	−0.40	−0.22	0.45 *	−0.44 *
MoCA	−0.30	−0.10	−0.21	1.00 **	−0.33	−0.16	−0.19	−0.53 **	−0.57 **	−0.39	−0.63 **	−0.49 *	−0.59 **	−0.22	0.51 *	−0.02	0.28	−0.60 **	0.54 **
UPDRS III	0.30	0.35	0.51 *	−0.33	1.00 **	0.12	0.35	0.61 **	0.70 **	0.55 **	0.69 **	0.54 **	0.68 **	0.42	−0.46 *	−0.32	−0.37	0.66 **	−0.69 **
LEDD	−0.00	0.40	0.13	−0.16	0.12	1.00 **	0.35	0.25	0.24	0.34	0.24	0.21	0.26	0.29	−0.27	−0.33	−0.22	0.15	−0.12
FOG-Q	−0.17	0.08	0.29	−0.19	0.35	0.35	1.00 **	0.45 *	0.43 *	0.57 **	0.41	0.35	0.45 *	0.44 *	−0.54 **	−0.52 *	0.06	0.36	−0.26
Stand Up Time	0.21	0.25	0.30	−0.53 **	0.61 **	0.25	0.45 *	1.00**	0.93 **	0.86 **	0.92 **	0.74 **	0.95 **	0.61 **	−0.67 **	−0.34	−0.24	0.84 **	−0.77 **
Walk Away Time	0.28	0.29	0.33	−0.57 **	0.70 **	0.24	0.43 *	0.93**	1.00 **	0.88 **	0.99 **	0.75 **	0.99 **	0.66 **	−0.71 **	−0.34	−0.33	0.93 **	−0.90 **
Rotation Time	0.22	0.31	0.32	−0.39	0.55 **	0.34	0.57 **	0.86**	0.88 **	1.00 **	0.82 **	0.77 **	0.90 **	0.85 **	−0.67 **	−0.38	−0.27	0.76 **	−0.72 **
Walk Back Time	0.30	0.31	0.34	−0.63 **	0.69 **	0.24	0.41	0.92**	0.99 **	0.82 **	1.00 **	0.73 **	0.98 **	0.62 **	−0.74 **	−0.35	−0.35	0.95 **	−0.91 **
Sit Down Time	0.27	0.22	0.19	−0.49 *	0.54 **	0.21	0.35	0.74**	0.75 **	0.77 **	0.73 **	1.00 **	0.81 **	0.58 **	−0.68 **	−0.18	−0.32	0.59 **	−0.57 **
Total Time	0.29	0.30	0.33	−0.59 **	0.68 **	0.26	0.45 *	0.95**	0.99 **	0.90 **	0.98 **	0.81 **	1.00 **	0.68 **	−0.74 **	−0.34	−0.34	0.91 **	−0.87 **
Rotation steps	0.24	0.38	0.42	−0.22	0.42	0.29	0.44 *	0.61**	0.66 **	0.85**	0.62**	0.58 **	0.68 **	1.00 **	−0.51 *	−0.27	−0.33	0.65 **	−0.64 **
Step correlation	−0.21	−0.17	−0.39	0.51 *	−0.46 *	−0.27	−0.54 **	−0.67 **	−0.71**	−0.67 **	−0.74 **	−0.68 **	−0.74 **	−0.51 *	1.00 **	0.41	0.20	−0.66 **	0.62 **
Step frequency	0.25	−0.26	−0.40	−0.02	−0.32	−0.33	−0.52 *	−0.34	−0.34	−0.38	−0.35	−0.18	−0.34	−0.27	0.41	1.00 **	0.08	−0.35	0.28
Sway	−0.38	−0.42	−0.22	0.28	−0.37	−0.22	0.06	−0.24	−0.33	−0.27	−0.35	−0.32	−0.34	−0.33	0.20	0.08	1.00 **	−0.40	0.45 *
Step number	0.35	0.31	0.45 *	−0.60 **	0.66 **	0.15	0.36	0.84 **	0.93**	0.76 **	0.95 **	0.59 **	0.91 **	0.65 **	−0.66 **	−0.35	−0.40	1.00 **	−0.97 **
Step length	−0.43	−0.28	−0.44 *	0.54 **	−0.69 **	−0.12	−0.26	−0.77 **	−0.90 **	−0.72 **	−0.91 **	−0.57 **	−0.87 **	−0.64 **	0.62 **	0.28	0.45 *	−0.97 **	1.00 **

FDR corrected * *p* < 0.05, ** *p* < 0.01; Stress Index, Baevsky’s Stress Index; FDR, False Discovery Rate; FOG-Q, Freezing Of Gait Questionnaire; HF, High frequency; HR, Heart Rate; HRV, Heart Rate Variability; LEDD, Levodopa Equivalent Daily Dosages; LF, Low frequency; MoCA, Montreal Cognitive Assessment; PD, Parkinson’s Disease; RMSSD, Root Mean Square of Successive Differences; RR, R-R interval; SDNN, Standard deviation of NN intervals; TUG, Timed up and go test; UPDRS, Unified Parkinson’s Disease Rating Scale.

## Data Availability

Data are available upon formal request by any identifiable researcher.

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
