# Peer review of "Sympathetic Burden Measured Through a Chest-Worn Sensor Correlates with Spatiotemporal Gait Performances and Global Cognition in Parkinson’s Disease"

_sensors, 2025, doi:10.3390/s25185756_

Round 1
Reviewer 1 Report
Comments and Suggestions for Authors
The manuscript addresses an interesting and clinically relevant topic, exploring the association between heart rate variability (HRV), gait parameters, and cognition in Parkinson's disease (PD) using a chest-worn sensor and smartphone-based gait analysis. The study is well written and structured, with a sound rationale supported by relevant literature. The combination of autonomic and gait metrics offers potential translational value for biomarker development in PD.
However, several issues should be addressed to strengthen the manuscript:
The study is limited by a small, single-centre cohort (n = 28) with a male predominance (75%). This imbalance may mask potential sex-specific differences in autonomic control and gait. The limitation is acknowledged, but a brief discussion on how this bias could influence the observed associations is warranted.
Although multiple comparisons were corrected using the FDR method, the results section could better highlight which associations remained significant after correction, especially in Figure 1 and Table 2. Providing exact p-values for key correlations would enhance transparency.
The choice to use only time-domain HRV measures should be more critically justified. Since frequency-domain measures have shown complementary value in PD-related postural control, the omission may underestimate the strength of autonomic–motor associations.
All testing was performed in the levodopa ON state, which could influence both gait and autonomic parameters. Although acknowledged as a limitation, the discussion should briefly address the implications for interpreting HRV–gait relationships.
The inverse relationship between Stress Index and MoCA scores is a novel finding that deserves more mechanistic discussion. For example, consider elaborating on whether this link could reflect shared neural substrates or a more global disease burden effect.
The correlation heatmaps (Figure 1) are informative, but the legends should indicate whether colour scale values represent correlation coefficients or p-values, and ensure readability for colour-blind readers.
Given the pilot nature of the study, it would be beneficial to outline a concrete plan for validation in larger, more diverse cohorts, potentially including longitudinal follow-up to assess predictive value for gait decline.
Author Response
The manuscript addresses an interesting and clinically relevant topic, exploring the association between heart rate variability (HRV), gait parameters, and cognition in Parkinson's disease (PD) using a chest-worn sensor and smartphone-based gait analysis. The study is well written and structured, with a sound rationale supported by relevant literature. The combination of autonomic and gait metrics offers potential translational value for biomarker development in PD.
Thanks for the thorough revision, for your time, for the constructive comments on our work. We appreciated.
However, several issues should be addressed to strengthen the manuscript:
Q1: The study is limited by a small, single-centre cohort (n = 28) with a male predominance (75%). This imbalance may mask potential sex-specific differences in autonomic control and gait. The limitation is acknowledged, but a brief discussion on how this bias could influence the observed associations is warranted.
R1: thanks for pointing out our main limitations for further clarification. They have been further discussed as suggested. This have been highlighted in yellow in the discussion section.
Q2: Although multiple comparisons were corrected using the FDR method, the results section could better highlight which associations remained significant after correction, especially in Figure 1 and Table 2. Providing exact p-values for key correlations would enhance transparency.
R2: we totally agree with your opinion. Result text has been updated and all exact p-values have been reported in the supplementary tables of annex 2.
Q3: The choice to use only time-domain HRV measures should be more critically justified. Since frequency-domain measures have shown complementary value in PD-related postural control, the omission may underestimate the strength of autonomic–motor associations.
R3: we totally agree with this comment, this has been further commented in methods and discussion section and highlighted in yellow.
Q4: All testing was performed in the levodopa ON state, which could influence both gait and autonomic parameters. Although acknowledged as a limitation, the discussion should briefly address the implications for interpreting HRV–gait relationships.
R4: We agree with this, and better addressed this limitation as suggested (yellow text in discussion).
Q5: The inverse relationship between Stress Index and MoCA scores is a novel finding that deserves more mechanistic discussion. For example, consider elaborating on whether this link could reflect shared neural substrates or a more global disease burden effect.
R5: thanks for this comment, this has been argument in discussion section and highlighted in yellow.
Q6: The correlation heatmaps (Figure 1) are informative, but the legends should indicate whether colour scale values represent correlation coefficients or p-values, and ensure readability for colour-blind readers.
R6: Intersections contain r values, this has been added in the figure caption. Thank you for suggesting this.
Q7: Given the pilot nature of the study, it would be beneficial to outline a concrete plan for validation in larger, more diverse cohorts, potentially including longitudinal follow-up to assess predictive value for gait decline.
R7: this was added to the discussion as suggested, than you for your thorough review.
Please note that, due to an editing error in the initial draft, HRV was actually recorded during a 2-minute resting state in the standing position, consistent with previous work on the topic already cited in our reference list (10.1016/j.parkreldis.2023.105476).
Reviewer 2 Report
Comments and Suggestions for Authors
In this study, the authors claim that their findings suggest that impaired autonomic regulation contributes to functional mobility deficits in Parkinson’s Disease (PD), supporting the role of heart rate variability (HRV) as a biomarker in motor assessment.
While the motivation of the study is relevant and of interest to the field, the manuscript would benefit from additional methodological details and the inclusion of raw data figures that demonstrate the highly significant correlations reported. These improvements are essential to ensure clarity and transparency in the analysis.
Major comments:
-The HRV stress index should be clearly defined and briefly explained in both the abstract and the introduction. Without such clarification, the manuscript will be challenging to follow for readers who are not experts in HRV analysis.
-A more detailed explanation of high and low stress index values, particularly concerning sympathetic activity, should be provided in the introduction to contextualize their physiological meaning.
-The total number of subjects included in the study should be mentioned in the first paragraph of the Methods section, even if exclusions were later applied (as noted in line 91). This would provide readers with a more straightforward overview of the study design from the outset.
-Acronyms used in Figure 1 and Table 2 should be defined in the legends to facilitate understanding.
-It would also be helpful to add a brief note in each legend explaining the rationale or the main result illustrated by the figure/table.
-The description of Figure 1 and Table 2 (lines 136–137) is too brief. A more detailed explanation should be added to clarify the significance and implications of these results.
-Graphs showing the correlations reported in Table 2 should be included. This would help readers verify that the comparisons were correctly analyzed and correspond to the reported results.
-A subsection in the Methods should be included to explain in detail how the correlations were obtained.
-The title refers to “global cognition,” but based on Figure 1 and Table 2, it is not clear which variables are being used to represent global cognition. Please clarify how “global cognition” was analyzed and ensure this is consistent throughout the manuscript.
Author Response
In this study, the authors claim that their findings suggest that impaired autonomic regulation contributes to functional mobility deficits in Parkinson’s Disease (PD), supporting the role of heart rate variability (HRV) as a biomarker in motor assessment.
While the motivation of the study is relevant and of interest to the field, the manuscript would benefit from additional methodological details and the inclusion of raw data figures that demonstrate the highly significant correlations reported. These improvements are essential to ensure clarity and transparency in the analysis.
Thankyou for the time spent on our work and for your insightful comments, that have been answered point by point below.
Major comments:
Q1: The HRV stress index should be clearly defined and briefly explained in both the abstract and the introduction. Without such clarification, the manuscript will be challenging to follow for readers who are not experts in HRV analysis.
R1: We agree with this and provided the requested updates in the abstract (briefly) and in the main text in introduction and methods. See changes highlighted in blue.
Q2: A more detailed explanation of high and low stress index values, particularly concerning sympathetic activity, should be provided in the introduction to contextualize their physiological meaning.
R2: We totally agree on this point and provided the requested updates in the introduction and methods. See changes highlighted in blue.
Q3: The total number of subjects included in the study should be mentioned in the first paragraph of the Methods section, even if exclusions were later applied (as noted in line 91). This would provide readers with a more straightforward overview of the study design from the outset.
R3: we understand your request and provided the mentioned changes.
Q4: Acronyms used in Figure 1 and Table 2 should be defined in the legends to facilitate understanding.
R3: Acronyms were added as suggested, thank you.
Q5: It would also be helpful to add a brief note in each legend explaining the rationale or the main result illustrated by the figure/table. Se text highlighted in blue.
R5: Thanks, we understand your request, and an explanatory note was added to each figure/table and/or in the result and discussion section. Se text highlighted in blue.
Q6: The description of Figure 1 and Table 2 (lines 136–137) is too brief. A more detailed explanation should be added to clarify the significance and implications of these results. Se text highlighted in blue.
R6: Agree, we apologize to have been to concise in this. The description of figure/tables was expanded as suggested. Se text highlighted in blue.
Q7: Graphs showing the correlations reported in Table 2 should be included. This would help readers verify that the comparisons were correctly analyzed and correspond to the reported results.
R7: Agree. This has been included in Annex 2. Thankyou.
Q8: A subsection in the Methods should be included to explain in detail how the correlations were obtained.
R8: This has been better described in methods.
Q9: The title refers to “global cognition,” but based on Figure 1 and Table 2, it is not clear which variables are being used to represent global cognition. Please clarify how “global cognition” was analyzed and ensure this is consistent throughout the manuscript.
R9: We apologize about this. Further specification on MoCA test were added throughout the manuscript.
Please note that, due to an editing error in the initial draft, HRV was actually recorded during a 2-minute resting state in the standing position, consistent with previous work on the topic already cited in our reference list (10.1016/j.parkreldis.2023.105476).
Reviewer 3 Report
Comments and Suggestions for Authors
The paper investigates the correlations between resting-state heart rate variability and gait parameters during comfortable and fast walking in patients with idiopathic Parkinson’s disease. The manuscript is well-structured and clearly written, making it easy to follow. However, the importance of the proposed work is not sufficiently emphasized. While the results may provide useful insights for clinical applications, the study does not introduce new sensing technology or propose novel methods for sensor data processing, which limits its contribution in the context of Sensors.
Major concerns:
- The authors need to better highlight how their findings advance the sensing field. For example, by discussing potential implications for sensor-based monitoring systems, novel applications in gait assessment, or integration into real-world health technologies.
- The Materials & Methods section does not describe the existing sensors to measure the HRV and spatiotemporal gait parameters in details. This limits the reader’s understanding of the accuracy and reliability of the measurements. More details on the specifications, validation, and limitations of the chosen devices are necessary. Without such information, concerns arise regarding the generalizability of the approach, since the reported results may depend strongly on the exact sensors and experimental setup.
- The Introduction does not contain sufficient information for the readers to understand the background and the importance of the proposed research, especially for the lack of the related works. A brief summary of the main contribution of this paper is also recommended.
Minor concerns:
- In Section 2.4, 'standard deviation' is defined user lower case letter for acronym.
- Please use upper letter for the first letter when referencing figures and tables.
- Some acronyms are defined multiple times across the paper. For example, HRV.
Author Response
Q1: The paper investigates the correlations between resting-state heart rate variability and gait parameters during comfortable and fast walking in patients with idiopathic Parkinson’s disease. The manuscript is well-structured and clearly written, making it easy to follow. However, the importance of the proposed work is not sufficiently emphasized. While the results may provide useful insights for clinical applications, the study does not introduce new sensing technology or propose novel methods for sensor data processing, which limits its contribution in the context of Sensors.
R1: Thank you for the positive overall evaluation of our work and for your insightful comments. Regarding the specific opinion regarding our contribution to Sensors, we are participating to a clinical special issue, accepting studies with similar approaches. I hope you can consider our revised work in the light of this crucial information, which if missing or not properly addressed can indeed condition your opinion about the suitability of the paper for the journal.
Major concerns:
Q2: The authors need to better highlight how their findings advance the sensing field. For example, by discussing potential implications for sensor-based monitoring systems, novel applications in gait assessment, or integration into real-world health technologies.
R2: this has been discussed as requested, see the paragraph highlighted in green in the discussion section
Q3: The Materials & Methods section does not describe the existing sensors to measure the HRV and spatiotemporal gait parameters in details. This limits the reader’s understanding of the accuracy and reliability of the measurements. More details on the specifications, validation, and limitations of the chosen devices are necessary. Without such information, concerns arise regarding the generalizability of the approach, since the reported results may depend strongly on the exact sensors and experimental setup.
R3: We totally agree with this comment, thanks for highlighting this. Method section has been updated with relevant informations about the sensing technology used. See paragraphs in green in the method section.
Q4: The Introduction does not contain sufficient information for the readers to understand the background and the importance of the proposed research, especially for the lack of the related works. A brief summary of the main contribution of this paper is also recommended.
R4: we understand your comment and fully agree with this. A brief summary of the main contribution of this paper has been added to the introduction as suggested (see text highlighted in green).
Please note that, due to an editing error in the initial draft, HRV was actually recorded during a 2-minute resting state in the standing position, consistent with previous work on the topic already cited in our reference list (10.1016/j.parkreldis.2023.105476).
Round 2
Reviewer 2 Report
Comments and Suggestions for Authors
The authors have satisfactorily responded to all my comments
Author Response
Thanks for your work on our manuscript-
Reviewer 3 Report
Comments and Suggestions for Authors
The revision addressed most of my previous comments. Here are some remaining concerns:
- In Section 2.4, “standard deviation” is defined using a lowercase letter for the acronym.
- Please use an uppercase letter for the first word when referencing figures and tables.
- The issue of multiple definitions of acronyms still exists, such as MSA.
- There is a redundant “Stress Index” in parentheses when defining SI on page 4.
- Please consider citing the URLs in the Reference section instead of placing the links directly in the main text.
- Figures and tables require significant improvement and reformatting. For example, the text in Figure 1 is very small. There are also blank pages after figures, or pages with only a few lines of text.
- Please be consistent on the reference citing. For example, line 141 should only cite it as [13] instead of [13Baevsky et al., 2009].
Author Response
Q1) In Section 2.4, “standard deviation” is defined using a lowercase letter for the acronym.
R1) thanks, this has been corrected
Q2) Please use an uppercase letter for the first word when referencing figures and tables.
R2) thanks, this has been corrected
Q3) The issue of multiple definitions of acronyms still exists, such as MSA.
R3) thanks, this has been corrected
Q4) There is a redundant “Stress Index” in parentheses when defining SI on page 4.
R4) thanks, this has been corrected
Q5) Please consider citing the URLs in the Reference section instead of placing the links directly in the main text.
R5) We have considered it, but prefer to keep the urls where they are. Thanks anyway
Q6) Figures and tables require significant improvement and reformatting. For example, the text in Figure 1 is very small. There are also blank pages after figures, or pages with only a few lines of text.
R6) thanks for this, we have now did a lot of work to fix what this reviewer suggested. Figure 1 has been improved in character dimension. All figures and tables were moved at the bottom of the ms. All future changes will be made with the help of the editorial assistance.
Q7) Please be consistent on the reference citing. For example, line 141 should only cite it as [13] instead of [13Baevsky et al., 2009].
R7) thanks, this has been corrected